

# Brief Communication: A nonlinear self-similar solution to barotropic flow over varying topography

Ruy Ibanez[1], Joseph Kuehl[2], Kalyan Shrestha[3], and William Anderson[3]

[1]Mechanical Engineering Department, University of Rochester, Rochester, NY 14627, United States
[2]Mechanical Engineering Department, University of Delaware, Newark, DE 19716, United States
[3]Mechanical Engineering Department, University of Texas Dallas, Dallas, TX 75080, United States

*Correspondence to:* Joseph Kuehl (jkuehl@udel.edu)

**Abstract.** Beginning from the Shallow Water Equations (SWE), a nonlinear self-similar analytic solution is derived for barotropic flow over varying topography. We study conditions relevant to the ocean slope where the flow is dominated by Earth's rotation and topography. The solution is found to extend the topographic $\beta$-plume solution of Kuehl (2014) in two ways: 1) The solution is valid for intensifying jets. 2) The influence of nonlinear advection is included. The SWE are scaled to the case of a topographically controlled jet, then solved by introducing a similarity variable, $\eta = cx^{n_x}y^{n_y}$. The nonlinear solution, valid for topographies $h = h_0 - \alpha xy^3$, takes the form of the Lambert W Function for pseudo velocity. The linear solution, valid for topographies $h = h_0 - \alpha xy^{-\gamma}$, takes the form of the Error Function for transport. Kuehl's results considered the case $-1 \le \gamma < 1$ which admits expanding jets, while the new result consider the case $\gamma < -1$ which admits intensifying jets and a nonlinear case with $\gamma = -3$.

10  *Copyright statement.* TEXT

## 1 Introduction

Slope topography represents both a barrier to large scale geophysical fluid transport as well as an important location of mesoscale feature generation. Standard quasi-geostrophic theory (Pedlosky 1987) indicates that large scale circulation features act in such a way as to conserve their potential vorticity, leading to the standard result of flow along (as opposed to across) topographic contours. Thus, slope topography creates a barrier between the open and coastal oceans, often inhibiting the transport of nutrient rich waters into the coastal zone and at the same time trapping pollutants in the coastal zone.

As both numerical and observational approaches have limitations with respect to modeling the slope region, the objective of this brief communication is to provide an analytic framework for flow along slope topographies. Such a framework will serve as an idealized backbone upon which observational, numerical, experimental and further theoretical work can build and provide a point of comparison for better interpretation of the respective dynamics. In particular, the results presented have implications for cross-topography exchange as well as provide significant insight into the coupling between the slope bottom boundary layer and interior water column dynamics.



## 2 Problem Formulation

The problem formulation considered in this work follows that of Zavala & van Heijst (2002), Kuehl (2014) and Kuehl & Sheremet (2014). A rotating, single fluid layer is considered which flows along a sloping bottom topography (ie along slope barotropic flow). The momentum equations and continuity equation (1) for this situation are:

$$u_t - (f+\omega)v = -(p-e)_x + \nu\nabla^2 u$$
$$v_t + (f+\omega)u = -(p-e)_y + \nu\nabla^2 v$$
$$h_t + (hu)_x + (hv)_y + \nabla\cdot\Pi_E = 0, \tag{1}$$

where $u, v$ are the across and along slope flow velocities, respectively, $h$ is the fluid depth, $p$ is the pressure anomaly divided by water density ($\rho$), $e$ is kinetic energy per unit mass, $\nu$ is the viscosity and $f$ is the Coriolis parameter. The effect of the

10 viscous bottom boundary layer is accounted for by a small correction term $\mathbf{\Pi}_E = -h_E\hat{k}\times\boldsymbol{u}$, the Ekman flux. Its divergence, $div(\mathbf{\Pi}_E) = -h_E\omega$, represents first-order Ekman suction at a solid boundary with Ekman layer depth, $h_E = \sqrt{2\nu/f}$. Taking the curl of momentum equations, defining the vorticity as $\omega = v_x - u_y$, defining a transport function $\psi$, where $\psi_x = hv$ and $\psi_y = -hu$, and simplifying by letting $q = \frac{f+\omega}{h}$, gives us the vorticity-transport equation (2),

$$\omega_t + J(\psi, q) = \nu\nabla^2\omega - \frac{h_e}{2}q\omega. \tag{2}$$

It is standard to expand the Jacobian, $J(\psi, q) = \frac{1}{h}(\psi, \omega) - \frac{f}{h^2}(\psi, \eta) + \frac{\beta^{(x)}}{h}\psi_y - \frac{\beta^{(y)}}{h}\psi_x$, where $\beta^{(x)} = (h_x f)/h$ and $\beta^{(y)} = (h_y f)/h$ are the average topographic beta-effects and $\eta$ is a small free-surface displacement.

Kuehl (2014) provide a scaling analysis which justified that Ekman dissipation is the dominant dissipative term and that relative vorticity is dominated by cross-stream shear, $\omega \approx \frac{1}{h}\psi_{xx}$. These assumptions are valid for flows which exhibit scale separation between the along and cross flow (topography) directions and are thus valid for flow along the oceanic slope. These

20 assumptions, along with the steady flow assumption, truncating a Taylor expansion in $\frac{1}{h}$ at leading order (neglecting terms of $O(\frac{1}{h^2})$), and assuming $f \gg \omega$ are again made and result in a leading order governing equation of the form:

$$\psi_x\psi_{xxy} - \psi_y\psi_{xxx} + fh_x\psi_y - fh_y\psi_x = -\frac{fh_E}{2}\psi_{xx}. \tag{3}$$

This equation (with appropriate boundary conditions) describes the linear and first-order nonlinear dynamics of a barotropic flow along the oceanic slope. It is upon this equation that several analytic solutions will be presented.

## 3 Linear Solutions

### 3.1 Expanding Jet

Kuehl (2014) considered the linear case of equation 3,

$$fh_x\psi_y - fh_y\psi_x = -\frac{fh_E}{2}\psi_{xx}. \tag{4}$$



Noting its similarity to the heat equation, which has been pointed out by others (in particular Csanady 1978), Kuehl attempted to find a similarity solution. The solution derivation will be sketched through here for completeness (details in Kuehl 2014). Assuming

- topography of the form $h = h_0 - \alpha x y^{-\gamma}$,

- similarity variable $\zeta = x \left( ky \right)^n$,

- boundary conditions $\psi(-\infty, y) = 0$ and $\psi(\infty, y) = Q$,

- initial condition $\psi(x, 0) = Q sgn(x)$,

equation 4 reduces to $-2\zeta g' = g''$, where $g = \psi/Q$, with conditions $n = -\frac{1+\gamma}{2}$ and $k = \left[ \frac{\alpha}{2h_E} \left(1 - \gamma\right) \right]^{\frac{1}{2n}}$. This equation has a well-know solution, $\psi = Q \left[ \frac{\text{erf}(\zeta) + 1}{2} \right]$, and parameters $\alpha$ and $\gamma$ may be set to mimic the desired topography. The "topographic

$\beta$-plume" solution is valid in the parameter range $-1 \leq \gamma < 1$. For the solution to be real, we must have $\gamma < 1$ and for $\gamma < -1$, the jet would be compressing, which does not satisfy the initial conditions. Physically, the Ekman pumping in the bottom boundary layer is relaxing the topographic vorticity control, allowing the jet to spread across isobaths.

## 3.2   Compressing Jet

In nature, compressing (or intensifying) jet are often observed and an analysis of ocean slope topography finds many locations

where $\gamma < -1$ is relevant (Ibanez 2016). To extend the above result to the case of compressing jets, the initial condition used above must be revisited. Similarity solutions require one point of reference to tether the solution. It is most common to place this singularity at the origin, as is done above and in many other classical cases such as the Blasius boundary layer (Blasius 1908, Rogers 1992). However, in the present case, we choose to relocate the singularity to $y = \infty$. Upon relocation, the solution given above is still valid but the domain of physical relevance of the solution has a slightly altered interpretation.

For the expanding jet case, the analytical solution is valid over the domain $y = [0 : \infty]$. However, the physical relevance of the solution demands the neglect of the region near $y = 0$, due to the singularity, as well as the region near $y = \infty$, as this region violates the across and along jet scale separation assumption. Though, the interior solution is indeed a physically relevant description of geophysical systems. For the compressing jet case, the situation is simply reversed. In this case, the analytical solution is still valid over the domain $y = [0 : \infty]$. However, the physical relevance of the solution demands the neglect of the

region near $y = 0$, as this region violates the across and along jet scale separation assumption, and the region near $y = \infty$, due to the singularity, but again the interior solution is a physically relevant description of geophysical systems. The region of applicability is ultimately governed by the assumption $\omega \approx \frac{1}{h} \psi_{xx}$ (ie $\psi_{xx} \gg \psi_{yy}$), which reasonable but should be check in each particular application.



## 4 Nonlinear Solution

Motivated by the success and utility of the linear solutions provided above, we seek a similarity solution for the nonlinear case (eqn. 3). Again, consider the normalized transport function, $g = \frac{\psi}{Q}$, and introduce a similarity variable of the form $\eta = cx^{n_x}y^{n_y}$, where $c, n_x, n_y$ are constants. The relevant derivatives take the form:

$$g_y = g'\frac{\partial \eta}{\partial y} = n_y cx^{n_x}y^{n_y-1}g'$$

$$g_x = g'\frac{\partial \eta}{\partial x} = n_x cx^{n_x-1}y^{n_y}g'$$

$$g_{xx} = \frac{\partial}{\partial x}\left[g'\frac{\partial \eta}{\partial x}\right] = g''\left(\frac{\partial \eta}{\partial x}\right)^2 + g'\frac{\partial^2 \eta}{\partial x^2}$$

$$= n_x^2 c^2 x^{2(n_x-1)}y^{2n_y}g'' + n_x(n_x-1)cx^{n_x-2}y^{n_y}g'$$

$$g_{xxx} = g'''\left(n_x^3 c^3 x^{3(n_x-1)}y^{3n_y}\right)$$

$$+ \ 3g''\left(n_x cx^{n_x-1}y^{n_y}\right)\left(n_x(n_x-1)cx^{n_x-2}y^{n_y}\right)$$

$$+ \ g'\left(n_x(n_x-1)(n_x-2)cx^{n_x-3}y^{n_y}\right)$$

$$g_{xxy} = g'''\left(n_y cx^{n_x}y^{n_y-1}\right)\left(c^2 n_x^2 x^{2(n_x-1)}y^{2n_y}\right)$$

$$+ \ g''\left(2n_x cx^{n_x-1}y^{n_y}\right)\left(n_x n_y cx^{n_x-1}y^{n_y-1}\right)$$

$$+ \ g''\left(n_y cx^{n_x}y^{n_y-1}\right)\left(n_x(n_x-1)x^{n_x-2}y^{n_y}\right)$$

$$+ \ g'\left(cn_x(n_x-1)n_y x^{n_x-2}y^{n_y-1}\right).$$

In this work, we are interested in straight slope topographies. Upon setting $n_x = 1$, it is seen that the nonlinear terms simplify significantly. Specifically, all $g'g'$ terms are set to zero. Also, it is found that the $g'g'''$ terms cancel. Thus, the only remaining nonlinear term is the $g'g''$ term, which in equation 3, takes the form $Q^2 g'g'' c^3 n_y y^{3n_y-1}$. Ultimately, equation 3 becomes

$$\underbrace{Qg'\left[h_x fcn_y xy^{n_y-1} - h_y fcy^{n_y}\right]}_{1}$$

$$+\underbrace{Qg''\left[\frac{fh_e}{2}c^2 y^{2n_y}\right]}_{2} + \underbrace{Q^2 g'g''\left[c^3 n_y y^{3n_y-1}\right]}_{3} = 0. \tag{5}$$

It is now convenient to address the $y$ dependences of the coefficients in terms 2 and 3 of equation 5. We require the $y$ dependency of terms 2 and 3 to balance, i.e. $2n_y = 3n_y - 1$, which gives the condition $n_y = 1$. Thus, the similarity variable has the form $\eta = cxy$. Apply this condition, and upon division by the coefficient of term 2, yields

$$\underbrace{\frac{2}{h_e c}g'\left[h_x xy^{-2} - h_y y^{-1}\right]}_{1} + \underbrace{g''}_{2} + \underbrace{\frac{2Qc}{fh_e}g'g''}_{3} = 0. \tag{6}$$





Next, the bracketed portion of equation 6 in term 1 is considered. Recall, $h = h_0 - \alpha x y^{-\gamma}$, $h_x = -\alpha y^{-\gamma}$ and $h_y = \alpha \gamma x y^{-\gamma-1}$. We anticipate that the $x$ must be absorbed into an $\eta$ term, so the bracketed terms become

$$-\frac{\alpha}{c}\eta y^{-\gamma-3}(1+\gamma). \tag{7}$$

The $y$ dependence is removed with the condition $\gamma = -3$ and the terms in 7 reduce to $2\frac{\alpha}{c}\eta$. It is then found that equation 6 reduces to

$$\frac{4\alpha}{h_e c^2}\eta g' + g'' + \frac{2Qc}{fh_e}g'g'' = 0. \tag{8}$$

Note as expected, the limit of equation 8, as $Q \to 0$, recovers the linear solutions provided above with $\frac{4\alpha}{h_e c^2} = 2$.

Thus, for topography of the form $h = h_0 - \alpha x y^3$ and a similarity variable of the form $\eta = cxy$, the nonlinear PDE, equation 3, reduces to a nonlinear ODE the form

$$\eta g' = -(K_1 + K_2 g')g'', \tag{9}$$

with $K_1 = \frac{h_e c^2}{4\alpha}$ and $K_2 = \frac{Qc^3}{2f\alpha}$.

Equation 9 can be solved for $g'$ by using separation of variables. Let $g'(\eta) = u(\eta)$ (the "sudo velocity") so $\eta u = -(K_1 + K_2 u)\frac{du}{d\eta}$ or $\eta d\eta = -(K_1 + K_2 u)\frac{du}{u}$. Integrating both sides yields

$$\frac{\eta^2}{2} + m = -(K_1 \ln u + K_2 u), \tag{10}$$

where $m$ is an integration constant related to the total transport.

It is possible to solve equation 10 for u, by using the Lambert W-Function ($W$),

$$u(\eta) = \frac{K_1 W\left(\frac{K_2}{K_1}e^{-\frac{2m+\eta^2}{2K_1}}\right)}{K_2}. \tag{11}$$

The integral of $u(\eta)$ is the analytic solution to the normalized transport equation, whose boundary conditions are $g(-\infty, y) = 0$, $g(\infty, y) = 1$ and $g(x, \infty) = Qsgn(x)$. However, the solution to the derivative of the transport function (pseudo velocity, $u$) is sufficient to calculate the flow field, as $\psi_x = Qg'(\eta)\frac{d\eta}{\partial x}$ and $\psi_y = Qg'(\eta)\frac{d\eta}{\partial y}$.

## 4.1 Calculation

It can be seen that $m$ is related to total transport by taking the analytic limit of equation 11 as $K_2 \to 0$ (which is an Error Function) and evaluating the transport boundary conditions. To complete the analytic solution in the nonlinear case, equation





11 can be integrated and an iterative method can be employed to determine $m$ based on the transport boundary condition. Alternatively, equation 9 can be solved numerically. A fourth order Runge-Kutta method coupled with a shooting algorithm was applied to iteratively meet the total transport boundary condition. It should be noted that the iterative numerical approach is based on a very small and sensitive velocity boundary condition, which cannot be taken at $-\infty$ but must be approximated at

a small finite value. In the linear and moderate nonlinear regime, the numerical and analytical solutions show good agreement (figure 1). However, as nonlinearity increases, the velocity boundary condition become extremely sensitive and difficult to iterate on. Thus, the great advantage of an analytical solution is that it is easily applicable at any amplitude.

## 5    Discussion

The solutions presented above are relevant to barotropic, along-slope flow over generic topographies of the form $h = h_0 -$

$\alpha x y^{-\gamma}$. For the linear solution cases, the Ekman pumping relaxes the topographic vorticity control via the bottom boundary layer. When $-1 \leq \gamma < 1$, the Ekman pumping out paces the topographic control and an expanding topographic $\beta$-plume solution is found. This represents cross-topographic transport due solely to bottom boundary layer processes. When $\gamma < -1$, the Ekman pumping is not able to overcome the topographic influence and a compressing topographic $\beta$-plume solution is found. Such compressing solutions result in intense currents, which may be subject to instability. For the special case, $h = h_0 - \alpha x y^3$,

a nonlinear solution is found. As seen in figure 1, the nonlinear solution broadens compared to the linear solution. At first this may seem to be a contradiction, however one must remember that in this case the topographic slope is rapidly increasing, with the influence to compress the jet. The influence of the nonlinear terms is to resist this compression. This is consistent with the expected tendency of flow inertia. The details of this nonlinear tendency are then relevant to the onset of barotropic instability (or other forms of instability, analysis of which is ongoing work). Note also that the nonlinear solution limits to the linear

solution (both analytically and numerically) as it must.

*Code and data availability.*   This is an analytical paper, the codes described are standard and easily reproduce from explanations provided in the text. Data availability is not applicable.

*Author contributions.*   Preparation of this manuscript was lead by Ibanez and Kuehl, however the ideas contained herein are the result of numerous discussion between all authors listed.

*Competing interests.*   No competing interests are present.

*Disclaimer.*   TEXT





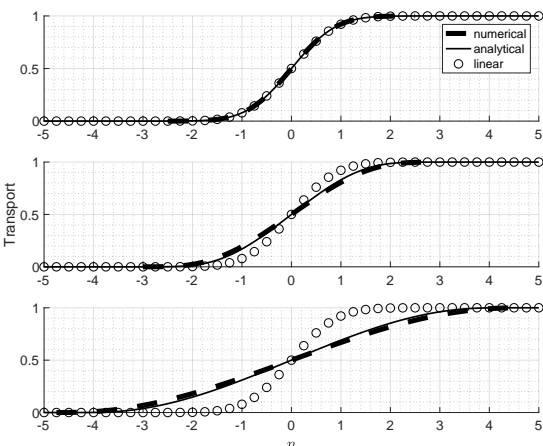

**Figure 1.** Comparison between linear (open circles), nonlinear numerical (thick dashed) and nonlinear analytic (solid lines) normalized transport functions. Plotted is the ratio $K_2/K_1$ (nonlinear coefficient over linear coefficient) of 0.001 (upper panel), 10 (middle panel) and 100 (lower panel) with $K_1 = 0.5$.

*Acknowledgements.* This work was supported by the Texas General Land Office, Oil Spill Program (Program Manager: Steve Buschang) under TGLO Contract #16-019-0009283 and the National Science Foundation, Physical Oceanography Program (Program Manager: Eric Itsweire) under grant #1657856.





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
