# Peer review of "Brief Communication: A nonlinear self-similar solution to barotropic flow over varying topography"

_Nonlinear Processes in Geophysics, 2017_

## Referee Comment (RC1) · Anonymous Referee #1 · 22 Dec 2017

*Review of the brief communication by Ibanez et al. entitled:*
**A nonlinear self-similar solution to barotropic flow over varying topography**

     This work focuses on finding analytical solutions for steady barotropic shallow-water flows over variable topography. Several key assumptions have been made: topography has special spatial dependence; flow goes only along the topography isolines; flow is steady; Rossby number is small; sea surface height has some other special spatial dependence. The manuscript is publishable, but its impact will be small, because the analytical solutions are way too special and, probably, not physical. Most likely, the obtained solutions are linearly unstable and, therefore, not reachable, and the relevant dynamics should be unsteady and full of topographic waves radiating from the large-scale flow and nonlinearly interacting with each other. Nevertheless, publications like this one are useful, as they help to maintain free of rust some classical mathematical approaches.

     More specific and minor comments are listed further below.

• Somewhere in the main text, can the concept of expanding and intensifying (contracting?) jets be made more explicit? Since the expansion/contraction would eventually lead to violations of the basic assumptions, it should be explicitly stated that the focus is on some kind of intermediate asymptotics. The same applies to the flow depth, which should formally outcrop somewhere.

• It seems that the sign in front of kinetic energy (by the way, it is not defined) in (1) should be minus.

*Page 2.* It should be highlighted somewhere, that the problem is formulated on an $f$-plane. What would happen in the case of a $\beta$-plane? Can the additional term be easily absorbed into topography? By the way, the reference to the $\beta$-plume is somewhat unclear and even awkward — please, explain better what is meant by it.

*Equation 2.* The variable $h_e$ should be $h_E$. There are other instances of this typo throughout. Line 15: Does the notation $(,)$ represent an undefined operator or are there missing '$J$'s? By my calculations, the 2nd term on the right-hand side, $f$ should be an $\omega$; please, check this out.

• The relationship between $\eta$ and $h$ is annoyingly never defined, although it becomes implicitly clear later on that $h$ is not the full depth, but the depth in the state of rest.

• Can it be explained, mathematically or physically, why $\gamma < -1$ induces "compressing" jets?

• Put equation (5) over one line. Why do you require the $y$-dependencies of terms 2 and 3 to balance?

*Page 5, line 12.* "pseudo velocity"?

• Discussion: Are there are any notable future steps that can be taken?

---

## Referee Comment (RC2) · Anonymous Referee #2 · 4 Jan 2018

This paper presents steps towards an approximate similarity solution to a form of the steady shallow-water equations that includes variable topography and a representation of Ekman pumping; this builds on earlier work by some of the authors. Eventual application to ocean flows is a stated goal, but no direct comparison is attempted here.

Unfortunately, I did not find the paper convincing, for two reasons. First, the assumed power law form of the topography seems arbitrary on physical grounds; I understand that it is needed for the similarity solution to work but that is just a mathematical reason with no oceanic relevance per se. Second, and more importantly to my mind, the derivation of the key governing equations (2) & (3) seems somewhat wrong (details below). If that is true then the similarity solution that follows later is hard to accept, of course. Perhaps this is just a confusion that could be cleared up under revision, but to

my mind a theoretical paper that is unclear on the foundations can serve little purpose. I regret that I cannot recommend this paper for publication.

Derivation steps leading to (2) & (3):

At first sight, (1) appear to be the standard shallow water equations with a free surface, augmented by Navier-Stokes frictional terms and another term representing an Ekman flux. The free surface seems to be there because of the time derivative in the height equation for h. However, closer inspection shows a pressure p in the momentum equations that is not linked to the height field h. The relationship between h and p is not discussed; as it stands (1) has more unknowns than equations.

My impression is that these are in fact the shallow water equations with a rigid lid on top, in which case p is the pressure just under the lid. But in that case the time derivative of h should not be there, so this is confusing to me.

Either way, the authors then take a curl, which makes the pressure term p disappear. They then assume that a mass transport stream function psi exists such that psi_x=hv and psi_y=-hu. This implies that the divergence of (hu,hv) is zero. This is again possible if a rigid lid is assumed such that h_t=0, but only if there is no Ekman pumping.

The authors then assume that the flow is steady, but because of the Ekman flux term in the continuity equation the divergence of (hu,hv) is NOT zero, so there cannot be such a stream function psi. It seems one can either have a stream function or the Ekman flux, but not both. But then the authors claim that (3), which approximates (2), should retain the Ekman term as a leading order dissipative term. To me, this looks inconsistent with the existence of a stream function psi in the first place, as indicated above.

---

## Author Comment (AC1) · 5 Jan 2018

We are thankful for the time and effort put into these thorough and thoughtful reviews. It would appear that during our efforts to condense this manuscript into a "brief communication," we have over simplified the presentation. Please allow us to clarify the comments of both reviewers.

1) Let us begin with the major criticism of reviewer 2 regarding the problem formulation. The issue is related to the physical origins of the Ekman suction term ($\vec{\Pi}_E = -h_E \hat{k} \times \vec{u}$) and our imprecise definition of transport function in the original manuscript. Kuehl and Sheremet (2014, JFM) contains the more complete treatment. The transport function

is defined through: $hu = \hat{k} \times \nabla\psi + \nabla\phi$, which is a generic decomposition. We over simplified this in our write-up to $hu = \hat{k} \times \nabla\psi$, which the review correctly identified. Now, $\nabla\phi = -\Pi_E$ which is from standard Ekman suction. Thus, we find $\nabla \cdot (hu) = \nabla^2\phi = -\nabla \cdot \Pi_E$, which is the steady state form of the continuity equation (eqn. 1). In the limit of no Ekman layer, the flow must follow isobaths. However, the presence of an Ekman layer allows the flow to cross isobaths. Also, the relationship between h and p is standard hydrostatic, but was not noted as we are considering the vorticity equation.

The text has been updated to clarify this by including the proper definition of $\psi$, which should address the reviewer concern. "defining an interior transport function $\psi$ through $hu = \hat{k} \times \nabla\psi + \nabla\phi$ (where $\nabla\phi = -\Pi_E$ represent Ekman divergence)"

2) Both reviewers have expressed concern over the "special" type of topography considered in this work, $h = h_0 - \alpha xy^{-\gamma}$. This is actually quite a generic topography and we feel this concern is not warranted. Consider the most simplistic model of the ocean slope, a constant slope $h = h_0 - \alpha x$. The next step is to allow that slope to vary linearly in the streamwise direction, $h = h_0 - \alpha xy$. Here we consider even more generic topographic variations, $h = h_0 - \alpha xy^{-\gamma}$. So the topography is not overly idealized, in the sense of a simple oceanic slope. It is a reasonable to ask how well the slope models the actual ocean topography. To this end, Ibanez developed a topography fitting code and has considered several oceanographic regions including the Norwegian coast, west Florida coast and west Keewenaw Peninsula in Lake Superior (Presented at: Ibanez et al. APS DFD 2016, Ibanez et al. APS DFD 2017 and in Ibanez's Masters Thesis), all of which exhibit large regions which fit the generic topography considered. Of particular interest is the Norwegian coast, in which linearly increasing topography seems to lead to eddy pinch off of the Norwegian coastal current. In the vicinity of the pinch-off, the topography appears to vary cubicly and thus the nonlinear solution may be applicable. The eddies shed from the Norwegian coast appear to be important for deep water formation and thus may have climatological importance. Also note, Kuehl

originally found good agreement between linearly varying topography on the northern slope of the Gulf of Mexico.

As this is a brief communication, analysis of ocean topography is not included. We have instead presented such analysis at APS DFD meetings and in prior publications. Thus, we feel the reviewers concerns about the topography are not warranted.

3) Reviewer 1 has expressed concern about the stability of the solution. Fully addressing this concern is on-going research via numerical linear stability analysis and rotating table experimental investigation. We can not fully answer this question at present, but can offer some motivation that the flow will be stable for some set of parameters. First, it is a common lab demonstration during geophysical fluid dynamics courses to illustrate Taylor columns over an isolated "bump" in topography. This is done with a sloping bottom on a rotating platform. I have run this demonstration and observed steady flow along sloping topography, indicating that stable solutions do exist. Also, the solution is evolving downstream, similar to the Blasius boundary layer, and preliminary stability analysis indicates a transition between stable and unstable regions of the flow. Finally, even in unstable regions of the flow, the steady analytic solution provides a means to study wave-mean-flow interaction and advective mixing processes.

We agree with the reviewer that the stability of the solution is an interesting question, but feel is does not diminish the relevance of the steady solution presented. As this is a brief communication, we feel a discussion of this point is not required in the text.

4) Reviewer 1 inquires about consideration of an f-plane versus a beta-plane. This solution considers a topographic beta-plane. It is assumed that the topographic beta terms (indicated between lines 10 and 15) are more significant than the planetary beta affect over the ocean slope. This is a reasonable assumption. Inclusion of a planetary beta-effect would contribute to the $\beta^{(x)}$ and $\beta^{(y)}$ terms.

5) Reviewer 1 has requested clarity concerning the "compressing" and "expanding" jet cases. The manuscript contains a discussion of the domain of relevance between lines 70 and 85. Compressing jets are those that approach a singularity at $y = \infty$, while expanding jets are those that start from a initial condition singularity at $y = 0$. Physically, this is determined by the exponentials ($n$ or $n_y$) in the similarity variable, which are related to $\gamma$.

We have included a clarifying statement at the end of section 3.2. "Thus, we have adopted the terminology that expanding jet are those with a singularity at the upstream source region ($y = 0$) and compressing jets as those with the singularity at downstream exit region ($y = \infty$)."

6) Pseudo velocity satisfies $g' = u$, while regular velocity comes from the definition of transport.

7) Requiring the y-dependencies balance is just part of the solution procedure. Perhaps more generic cases can be identified in the future.

8) Several typos and formatting issues were identified by the reviewers, which will be corrected. Thank you.

9) As this is a brief communication, we have chosen not to include a discussion of future work. However, we are currently working toward stability analysis of the solutions and rotating table experimental investigation.

Please also note the supplement to this comment:

https://www.nonlin-processes-geophys-discuss.net/npg-2017-66/npg-2017-66-AC1-supplement.pdf

---

## Editor Comment (EC1) · JM Restrepo (Editor) · 8 Jan 2018

There are a few items in the response to the referees that were not addressed fully:

Ref 1:

(1) define the kinetic energy and comment on the sign (Eq 1) (2) define the Jacobian operator (Eq 2) and reply to the ref concerning his/her question regarding the outcome, the RHS.

Ref 2:

(1) Although p (and e) will not play a role in the vorticity equation, it would be useful to be explicit in what these are. (2) There are some "cultural" issues with regard to the symbols and the names used for the various terms. For example, while there is no universally accepted nomenclature-to-equation pairing regarding the shallow water wave equation, the form used in the mss is different from the one that is familiar to a rather large community of nonlinear (as well as dispersive) wave researchers. I would suggest omission of (SWE) nomenclature. Along those lines, the use of $\eta$ for the similarity variable will clash with what the SWE community often uses for the displacement of the sea surface, away from the quiescent level. The use of the symbol is still ok, since there is a published history for this term in prior work, but perhaps the way to clarify this is to be more explicit about what $h$ is.

**A suggestion:**

The referees speculate that the solution may be special, i.e. the result of specialized balances (and/or unstable). This in itself is not a reason to not publish the result, but one thing that might be useful is to present the full scaling that leads to (Eq 1). This would be very useful in assuring the readers how the Taylor series works out and what terms are included/omitted. Alternatively, the authors can present a better review/explanation of the starting equation in the Zabala Sanson and van Heijst paper. The former alternative is more work but will be appreciated by the reader (and the authors). The latter would be expedient but still very useful.

---

## Author Response (AR1)

**Review Response 2: "Brief Communication: A nonlinear self-similar solution to barotropic flow over varying topography"**

Thank you for addition comments. We have clarified the manuscript, address the reviewers concerns, as described below.

1) We have included definitions of kinetic energy and the Jacobian.

2) The RHS of equation 2 has been verified against Zavala & van Heijst (2002).

3) The definitions of h (fluid depth of at rest state) has been clarified.

4) A statement has been added to clarify the use of $\eta$ as both surface displacement and similarity variable.

5) A statement that the "total water column depth is $h+\eta$" has been added.

6) The reader has been directed to Pedlosky's GFD book as well as Cushman-Roisin Intro to GFD book for background and scalings that lead to the shallow water equations (equation 1). The authors feel that as this is covered in well-know textbooks, it is not necessary to repeat here (particularly in a brief communication).

[revised manuscript text omitted]